# Investigating the Digestibility, Bioavailability and Utilization of Protein Blends in Older Adults Using a Dual Stable Isotope Tracer Technique

**DOI:** 10.3390/nu17213328

**Published:** 2025-10-23

**Authors:** Jake Cox, Bethan E. Phillips, James Bunce, Thomas Smart, Joshua Wall, Hannah Crossland, Daniel J. Wilkinson, Kenneth Smith, Philip J. Atherton

**Affiliations:** COMAP Research Group, School of Medicine, Faculty of Medicine and Health Science, University of Nottingham, Nottingham DE22 3DT, UK; mbyjc16@nottingham.ac.uk (J.C.); mbzbp@exmail.nottingham.ac.uk (B.E.P.); james.bunce@nhs.net (J.B.); thomas.smart2@nhs.net (T.S.); joshua.wall@nhs.net (J.W.); msahc19@exmail.nottingham.ac.uk (H.C.); mdzdw@exmail.nottingham.ac.uk (D.J.W.); mbzks@exmail.nottingham.ac.uk (K.S.)

**Keywords:** protein digestibility, dual stable isotope, skeletal muscle, protein synthesis, ageing

## Abstract

Objectives: The impact of combining animal and plant protein sources on digestibility is unclear, despite their increasing clinical use. Using a non-invasive dual stable isotope tracer approach, we assessed the digestibility, bioavailability and utilization of distinct protein blends in older adults, and associated plasma amino acid profiles and muscle protein synthesis (MPS) rates. Methods: Thirty-two older men (69 ± 3 y) consumed one of four protein blends (A (51:49, casein/soy); B and C (35:25:20:20, whey/casein/soy/pea); D (80:20, casein/whey)) alongside primed constant infusions of [1,2-^13^C_2_] leucine for 8 h. Arterialized blood and vastus lateralis muscle biopsies were collected during a trickle feed protocol with all blends providing 20 g total protein, universally labeled ^13^C-spirulina, and ^2^H-cell free amino acid mix to determine digestibility. This trial was registered at Clinicaltrials.gov (ID-NCT07038655). Results: No differences (^13^C:^2^H ratios) were found in digestibility between the protein blends (*p* > 0.05). Mean (±SEM) fed state MPS at 2.5 h was 0.078 ± 0.009%/h, 0.075 ± 0.012%/h, 0.085 ± 0.007%/h and 0.065 ± 0.011%/h for drinks A, B, C and D, respectively, with a main time effect observed (*p* < 0.01), but no significant differences between drinks. Plasma essential amino acids (EAAs) increased significantly from baseline for all blends by 40 min (*p* < 0.05), with no differences between blends at any time point. Conclusions: These findings suggest that protein quantity (and/or leucine content), rather than composition, appears to be the most important factor driving MPS. Future work should focus on clinical populations where protein requirements and digestibility characteristics may differ.

## 1. Introduction

Skeletal muscle mass is maintained through the dynamic equilibrium between muscle protein synthesis (MPS) and muscle protein breakdown (MPB), the balance of which dictates protein turnover. In healthy individuals, a positive net protein balance is predominantly driven by increases in MPS [1], which are of a much greater magnitude than the suppression of MPB [2,3]. The provision of amino acids through protein feeding is a key anabolic stimulus, resulting in a transient, approximate two- to threefold increase in MPS from basal levels in young healthy individuals [4]. Of these, the essential amino acids (EAAs), which are not synthesized within the body and can therefore only be obtained from dietary protein ingestion, are the most potent for stimulating MPS, even at relatively low doses [5]. In particular, the branched chain amino acid, leucine, has been shown to have a regulatory capacity in MPS via the activation of protein phosphorylation in the mTOR signaling pathway [6].

Ageing is typically accompanied by a progressive loss of muscle mass and function, termed sarcopenia [7]. This can be attributed to the blunted response to anabolic stimuli such as protein feeding and exercise that has been demonstrated in older individuals compared to young [8,9]. One potential approach to overcome this age-associated anabolic resistance would simply be to eat more protein throughout the day. The general consensus is that the current recommended dietary allowance of 0.8 g/kg/day of protein is inadequate for older individuals, and that this should be increased to 1.2–1.5 g/kg/day, with protein intake evenly distributed across three to four meals [10,11,12,13,14,15]. However, in a free-living environment, there are numerous practical limitations that cause older individuals to consume suboptimal quantities of protein. These include reduced energy demands decreasing total food intake, combined with the satiating effect of protein feeding, physical dependence and food insecurity limiting access to high-quality protein sources, other underlying conditions such as poor dentition and dysphagia, and potential shifts in food preference away from high protein meals [16]. Therefore, if simply increasing the quantity of protein consumed is not always practical, it is important to consider other factors such as the timing and distribution of protein feeds [12,17], the quality of the protein source [18,19], and interactions with additional factors such as exercise [20,21] and health status [22], to optimize protein intake [23]. Of these variables, protein quality has been at the forefront of recent research due to developments in the technical capacity to quantify this. Protein quality is determined by both the digestibility and amino acid composition of a protein source, which then dictates the amino acid bioavailability—the appearance in the systemic circulation—for uptake and utilization by muscle and other tissues and organs.

Historically, the quantification of protein digestibility in humans has required ileostomized participants or the invasive sampling of the terminal ileum through naso-ileal intubation [24]. However, the development of a minimally invasive dual stable isotope tracer approach to quantify protein digestibility facilitates application to vulnerable clinical and elderly populations [25]. Using this technique, the digestibility of various amino acids across a range of protein sources can be assessed, along with quantifying the effects of different strategies to enhance protein digestibility. Importantly, the minimally invasive nature of this technique facilitates the quantification of digestibility directly in the target population, such as older individuals who may be most in need of optimizing the bioavailability of amino acids from dietary sources.

To date, the majority of studies using the dual stable isotope tracer technique have focused on the validation of the technique [26,27], strategies to enhance the digestibility of otherwise poorly digested protein sources [28,29], or the digestibility quantification of commonly consumed animal and plant protein sources [30,31,32,33,34,35], with only one study applying the technique to a clinical population [36]. However, there is a lack of consideration among these studies of the impacts of protein digestibility on subsequent changes in plasma amino acid bioavailability and MPS. Considering these outcomes alongside digestibility is important, as they reflect the anabolic potential of the protein source in the target population and can elucidate the extent to which protein digestibility either improves or limits the quality of a protein source. Typically, the measurement of protein digestibility using a dual stable isotope approach requires an intrinsically labeled protein source provided alongside a differently labeled protein source of known digestibility as a reference. In the present study, we applied this approach to four unlabeled protein blends of differing compositions, with the aim of using the intrinsically labeled protein source as a reporter protein in conjunction with the labeled free amino acid mixture, which is assumed to represent 100% bioavailability. Using this approach, we assessed the bioavailability of the protein blends based on the relative dilution of the ^13^C:^2^H enrichment ratio in the plasma compared to the same ratio determined in the respective drinks, alongside measurements of plasma amino acid concentrations and MPS rates.

## 2. Methods

### 2.1. Participants

Thirty-two older men (mean ± SD age—69 ± 3 years, BMI—25.6 ± 3.0) volunteered to take part in the study. All participants provided written informed consent. Prior to their enrolment, all participants completed a comprehensive clinical examination and metabolic screening including liver function tests (LFT), thyroid function tests (TFT) and full blood count (FBC) at the School of Medicine, Royal Derby Hospital Centre. All participants were recreationally active. Participants with metabolic disease, respiratory disease, acute cerebrovascular or cardiovascular disease, renal disease, active inflammatory bowel disease, musculoskeletal or neurological disorders or malignancy within the previous five years were excluded.

### 2.2. Study Design

This study was conducted via a parallel group study design where each participant was randomly assigned to receive one of four distinct protein blend drinks, for a total of eight participants per drink. Participants were studied following an overnight fast and refrained from heavy exercise for 72 h before the study to limit any effect of prior exercise on MPS measures or any currently unknown interaction between exercise and protein digestibility. Participants were also instructed to follow their normal habitual diet in the days prior to their study visit to avoid any impact of altering the diet, particularly changes in protein intake that might impact nitrogen balance and protein turnover. Participants arrived at 08:00 on the morning of the study and had a cannula inserted into the antecubital vein of one arm for stable isotope tracer infusion, and a retrograde cannula inserted into a dorsal vein of the other arm for arterialized blood sampling using a hotbox. A primed, continuous infusion (0.7 mg·kg^−1^ prime, 1.0 mg·kg^−1^·h^−1^ continuous infusion) of [1,2-^13^C_2_] leucine was maintained throughout the study. Muscle biopsies were taken 1 h and 3 h after the onset of tracer infusion for the determination of basal MPS rates.

Participants then began a trickle-feed protocol for their randomized protein feed. The protein feeds each consisted of a blend of protein sources, as follows: drink A (51:49, casein:soy), drink B (35:25:20:20, whey:casein:soy:pea), drink C (35:25:20:20, whey:casein:soy:pea) and drink D (80:20, casein:whey). Notably, drinks B and C contained the same proportions of whey, casein, soy and pea protein, despite being distinct commercially available products. The comparison of these drinks would therefore serve as an internal control to check for the accuracy of the dual stable isotope tracer approach used in this study. The drinks were matched to provide 20 g of protein in total, and contained 900 mg U-^13^C spirulina and 300 mg U-^2^H free amino acids for the quantification of protein digestibility and a top up of [1,2-^13^C_2_] leucine equal to 8% of the leucine content of the drinks to maintain steady state enrichment during the feeding phase. Drinks were then topped up to equal volumes with water to maintain blinding. The protein drinks were divided into 17 aliquots. Three were given at the start of the feeding period to prime the pool, and one aliquot was given every 20 min thereafter. Further muscle biopsies were taken 2.5 h and 5 h after the first protein feed. Muscle biopsies were collected from the medial vastus lateralis using the conchotome technique under sterile conditions with 1% lidocaine as the local anesthetic [37]. Muscle tissue was washed in ice-cold phosphate-buffered saline, blotted dry and snap-frozen in liquid nitrogen, before being stored at −80 °C until analysis. Arterialized blood samples were collected at the time points outlined in Figure 1.

### 2.3. Measurement of MPS

Myofibrillar proteins were isolated, hydrolyzed and derivatized in accordance with our previously described techniques [38]. Briefly, 20–30 mg of muscle tissue was homogenized in ice-cold homogenization buffer (50 mM Tris–HCl (pH 7.4), 50 mM NaF, 10 mM β-glycerophosphate disodium salt, 1 mM EDTA, 1 mM EGTA, 1 mM activated Na_3_VO_4_ (all from Sigma-Aldrich, Poole, UK) and a complete protease inhibitor cocktail tablet (Roche, West Sussex, UK) at 10 µL·µg^−1^ of tissue. Homogenates were rotated for 10 min on a Vibrax shaker at a speed of 1500 rpm, and the supernatant was collected by centrifugation at 11,000× *g* for 15 min at 4 °C. The resulting pellet was washed twice with homogenization buffer to remove any excess sarcoplasmic protein and free amino acids. The pellet was then solubilized in 0.3 M NaOH to aid the separation of the soluble myofibrillar fraction from the insoluble collagen fraction by centrifugation. The myofibrillar fraction was then precipitated using 1 M perchloric acid (PCA) and pelleted by centrifugation before being washed twice with 70% ethanol. Protein-bound amino acids were released by acid hydrolysis using 0.1 M HCl and 1 mL dowex ion-exchange resin (50W-X8-200) overnight at 110 °C. Amino acids were purified by ion-exchange chromatography on dowex H^+^ resin columns before being eluted with NH_4_OH and derivatized. Samples were analyzed using gas chromatography–combustion–isotope ratio mass spectrometry and the fractional synthetic rate (FSR) of the myofibrillar proteins was calculated using the below precursor product equation:$$FSR\left(\%/h\right)=\frac{∆Em}{Ep}\times \frac{1}{t}\times 100$$

Here, ∆Em is the change in the [1,2-^13^C_2_] leucine enrichment between subsequent biopsies, *t* is the time interval between subsequent biopsies and *Ep* is the mean enrichment over the same period of the precursor for protein synthesis, where venous plasma α-ketoisocaproate (KIC) is used as a proxy for leucyl-tRNA, the immediate precursor for protein synthesis [39]. Briefly, plasma samples were deproteinized with 1 mL ice-cold ethanol and dried down, before preparing the KIC quinoxalinol-t-BDMS derivative for gas chromatography–mass spectrometry (GC-MS) analysis.

### 2.4. Measurement of Protein Digestibility

Arterialized plasma (200 µL) was deproteinized with 1 mL ice-cold ethanol and centrifuged at 18,000× *g* for five minutes. The supernatant was then collected and evaporated under nitrogen at 90 °C, followed by re-suspension in 500 µL 0.5 M HCl. Ethyl acetate (2 mL) was added to the samples and vortexed, with the top layer then discarded to remove lipids. The aqueous layer containing amino acids was evaporated at 90 °C under nitrogen and then resuspended in 75 µL 1% NaOH, 47 µL 1-propanol, 18 µL pyridine, 15 µL iso-octane and 15 µL propyl chloroformate. Samples were then transferred to an Eppendorf containing chloroform with the subsequent addition of 1 M HCl. Following vortex mixing, organic and aqueous phases were allowed to separate, and the organic layer was taken up and evaporated under nitrogen at 40 °C. The dried sample was then resuspended in hexane, ready for GC-combustion or pyrolysis-IRMS analysis to measure the ^13^C and ^2^H enrichments of amino acids.

Aliquots of each of the prepared protein-tracer drinks were hydrolyzed overnight in 0.1 M HCl and dowex slurry at 110 °C to release amino acids, which were then purified using dowex H^+^ resin columns, to provide baseline enrichment ratios for ^13^C- and ^2^H-labeled amino acids. Samples were then derivatized using the propyl chloroformate method described above.

Protein digestibility was determined based on the dilution of the plasma ^13^C-phe (APE)/^2^H-phe (APE) relative to the same ^13^C/^2^H ratio measured in the protein tracer drink.

For the calculation of plasma amino acid concentrations, norleucine internal standard was added to plasma samples prior to deproteinization, the propyl chloroformate derivative was prepared as described above, and concentrations were determined in reference to an amino acid standard curve of a known concentration.

### 2.5. Statistical Analysis

All data are presented as mean ± SEM, unless stated otherwise. Statistical analysis was performed using GraphPad Prism Version 10.3.1 (GraphPad, San Diego, CA, USA) and significance was defined as *p* < 0.05. Statistical comparisons between groups and time required for changes in plasma amino acid responses and MPS were carried out using two-way analysis of variance (ANOVA) with post-hoc Tukey’s testing. For the analysis of protein digestibility values and plasma amino acid AUC’s, one-way ANOVA was used with post-hoc Tukey’s testing.

## 3. Results

### 3.1. Plasma Amino Acid Concentrations

Following protein feeding, there was a significant increase in plasma EAA concentrations relative to baseline for all drinks, though this occurred by 20 min for drink D and by 40 min for drinks A, B and C (all *p* < 0.05), as demonstrated in Figure 1. These differences in the timings of the plasma EAA response did not result in any significant differences between drinks at any time points when performing multiple comparisons (all *p* > 0.05). There were also no significant differences in the AUC for plasma EAA concentrations between any of the drinks (all *p* > 0.05).

**Figure 1 nutrients-17-03328-f001:**
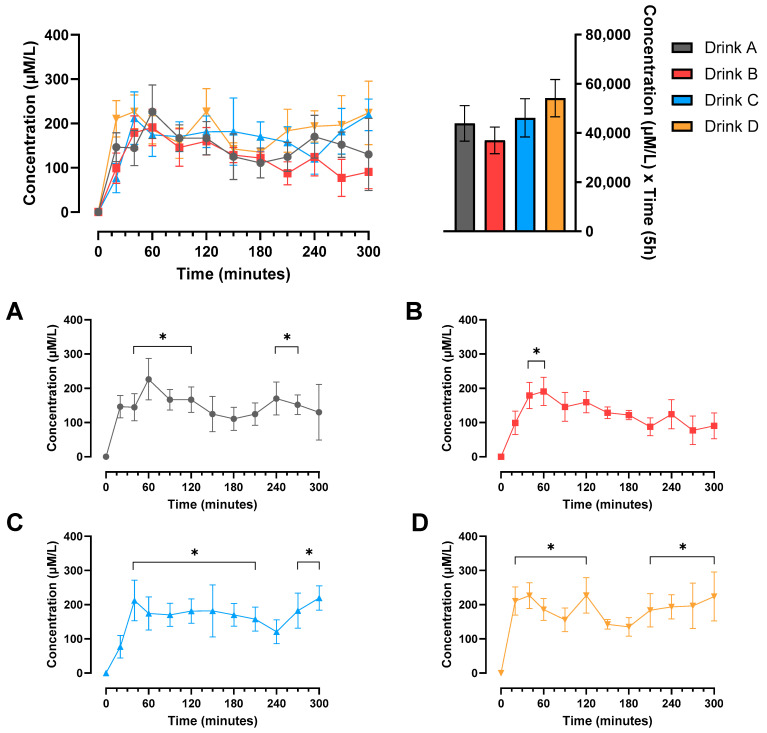
Change in plasma EAA concentrations compared to baseline over time and area under the curve (AUC) in older males. Values are mean ± SEM. Significant main treatment (*p* < 0.01) and time (*p* < 0.0001) effects observed. (**A**–**D**) The changes in plasma EAA concentrations compared to baseline in drinks A, B, C and D, respectively. * Significant difference from baseline at the time point (*p* < 0.05).

Changes in plasma leucine concentrations from baseline are shown in Figure 2. The plasma leucine concentrations increased significantly from baseline for all drinks, occurring by 20 min for drink D and by 40 min for drinks A, B and C (all *p* < 0.05). This more rapid response for drink D resulted in a significant difference between drinks D and C at 20 min (*p* < 0.05). There was also a significant difference between drink D and drink A at 300 min. There was no significant difference between drinks in the AUC for plasma leucine concentrations (all *p* > 0.05).

### 3.2. Protein Digestibility

Changes in the ^13^C:^2^H enrichment ratio in the plasma relative to the ^13^C:^2^H ratio in the drink will reflect changes in the relative bioavailability of the amino acids from the unlabeled protein and the ^13^C-labeled amino acids derived from the spirulina, assuming the ^2^H-labeled free amino acids represent 100% bioavailability. The greater dilution of the ^13^C label from the spirulina protein than the ^2^H label from the free amino acids indicates the greater digestibility and bioavailability of amino acids from the unlabeled protein blend. Conversely, if the ^13^C:^2^H ratio increased in the plasma, this would indicate that the U-^13^C spirulina was digested to a greater extent than the unlabeled protein blend. Mean ± SD protein ^13^C:^2^H ratio curves for phenylalanine are shown in Figure 3. The dotted lines represent the ^13^C:^2^H enrichment ratios in the respective drinks, and the solid lines show the change in ^13^C:^2^H enrichment ratios in the plasma over time, which is representative of the digestibility and bioavailability of the spirulina reporter protein in relation to the unlabeled protein blend. There were no significant differences in the plasma enrichment ratios between any drinks at any time points after reaching the plateau.

### 3.3. Myofibrillar FSR Response

Basal and 150 and 300 min postprandial myofibrillar protein synthesis rates in response to all four protein blends are shown in Figure 4. A significant main time effect for MPS was observed across the study (*p* < 0.01). No significant differences in mean FSR rates from baseline to 150 min were observed for any of the drinks (0.056 ± 0.007 vs. 0.078 ± 0.010, 0.056 ± 0.012 vs. 0.075 ± 0.012, 0.057 ± 0.013 vs. 0.085 ± 0.007 and 0.048 ± 0.012 vs. 0.065 ± 0.011 for drinks A, B, C and D, respectively, all *p* > 0.05). Within-group effect sizes (Cohen’s d) for the change from baseline to 150 min were strong (drink A, d = 0.83, 95% CI [−0.24, 1.90]; drink B, d = 0.80, 95% CI [−0.26, 1.86]; drink C, d = 0.88, 95% CI [−0.58, 2.34]; drink D, d = 0.89, 95% CI [−0.57, 2.36]). There were also no significant differences in MPS rates between drinks at any given timepoints (all *p* > 0.05).

## 4. Discussion

The quantification of protein digestibility using dual stable isotope tracers has been the focus of recent research due to the relatively minimally invasive nature of this technique compared to naso-ileal sampling, which was historically used to measure digestibility. Given that most of the research to date has aimed to validate the accuracy of the dual tracer technique or quantify the digestibility of commonly consumed protein sources, this study is the first to apply a dual stable isotope tracer technique to assess protein digestibility in older adults—a population in which it would be challenging to use the invasive mode of naso-ileal sampling. We have also extended this by including measures of MPS rates, providing us a full complement of data from the digestion of the protein sources to the subsequent bioavailability of amino acids, and then their eventual utilization at the muscle.

We observed no differences between drinks in the plasma phenylalanine ^13^C:^2^H enrichment ratios at plateau. This indicates that the ^13^C phenylalanine deriving from the spirulina reporter protein is diluted to the same extent with all four protein blends, meaning that the digestibility and bioavailability of these blends are equivalent. It is perhaps unsurprising that the bioavailability of all four protein sources was similar, given that they all consisted of high-quality proteins, namely, whey, casein, soy and pea proteins. Previous calculations of the standardized ileal digestibility of whey protein isolate, pea protein concentrate and soy protein isolate have been carried out in pigs, with values for phenylalanine digestibility reported at 98%, 92% and 96%, respectively [40]. Similarly, values for the phenylalanine digestibility of casein protein have been reported in humans using the naso-ileal intubation technique, with a digestibility score of 99.2% [41]. It is clear from the relative dilution of the ^13^C enrichment from the spirulina protein in our study that all four protein blends were highly digestible, more so than the spirulina protein itself, and that combining different protein blends does not negatively impact on the bioavailability of amino acids.

To date, studies assessing the digestibility of protein sources using the dual stable isotope tracer technique have utilized an intrinsically labeled protein source alongside a differently labeled reference protein or amino acid mixture. Universally labeled U-^13^C spirulina protein is commonly used as a reference protein to study digestibility, as it is freely available and relatively inexpensive when compared to other intrinsically labeled proteins. The use of spirulina has been validated, with mean spirulina protein digestibility for all EAAs being reported at 85.2% and ranging from 77.5% (lysine) to 95.3% (phenylalanine) [42]. In this study, we have used spirulina as a reporter protein, proposing that the relative dilution of the ^13^C label from the spirulina by the amino acids from an unlabeled test protein can be used to assess the relative digestibility of any unlabeled protein source or mixture. This is achieved using the ^13^C:^2^H ratio from the spirulina to the free amino acid mix as an index of the relative digestibility, i.e., a dilution of the ^13^C:^2^H ratio implies the greater digestibility of the protein under test when compared to spirulina. The advantage of this is that we can assess the digestibility of a range of protein blends and sources without the need to generate intrinsically labeled test proteins, which is relatively expensive and time consuming. However, a current limitation of this method is that further validation is required to determine the applicability and accuracy of the approach to measure protein digestibility, as the present study only used the technique to assess a range of unlabeled protein blends, which did not have differences in digestibility profiles. Additionally, further work is needed to develop the technique to be able to provide a value of true digestibility for unlabeled test protein sources; this means adopting the same approach but using intrinsically labeled proteins.

As we observed no differences in the digestibility of the drinks for phenylalanine, it is perhaps unsurprising that there were only very subtle differences in the time course of plasma EAA and leucine concentration responses, with a significant difference only observed between drinks D and C at 20 min for EAAs. In addition to this, there was a notable rapid response to drink D for both EAA and leucine plasma concentrations, whereby the increase from baseline reached significance at 20 min for drink D but not for drinks A, B and C. These findings are perhaps somewhat surprising given that drink D was predominantly made up of casein protein, which is a slowly digested protein that has been found to induce a lower early phase plasma EAA response that is more prolonged than the rapid, transient response to whey protein [43]. Indeed, this slower digestive profile is reflected somewhat in Figure 3, where we can observe that drinks A and D, with high casein contents of 51% and 80%, respectively, did not reach plateau until approximately 240 min, whereas drinks B and D, each containing only 25% casein, plateaued at approximately 150 min. Therefore, a potential explanation for the more rapid early plasma EAA increase observed in response to drink D despite the higher casein content could relate to the amino acid composition of the protein blends. Though we did not measure the amino acid composition of the blends in this study, it has been well established that plant proteins such as soy and pea have a lower EAA content than animal protein sources such as whey and casein [44,45]. Given that drink D contained only animal proteins whereas drinks A, B and C all contained at least 40% plant protein, it could be that there was a higher EAA content in the protein of drink D, which was sufficient to overcome the slower digestibility of casein protein in the first twenty minutes post feeding. Another potential factor that may have contributed to this unexpected finding was that we utilized a trickle-feed protocol for the measurement of protein digestibility. Casein is typically considered a slower digesting protein as it precipitates in the acidic environment of the stomach and coagulates in the gut milieu, slowing the digestion and hydrolysis of proteins. However, a trickle feed approach may not cause this to slow to the same extent, as there is a much smaller quantity of protein coagulating in the gut compared to when it is fed as a single bolus.

Importantly, the lack of significant differences in MPS rates between any drinks at 150 min reflects the similar patterns observed for drink digestibility and amino acid bioavailability, where all four drinks performed relatively similarly. It is also possible that the lack of significant differences from baseline to 150 min in any of the drinks reflects the blunted anabolic response typically seen in ageing. This phenomenon, whereby increases in MPS rates following anabolic stimuli in older individuals are reduced compared to younger individuals, has been well documented [11,46,47]. Other contributing factors to the lack of changes observed in MPS in this study may include the relatively low sample size of eight participants per group, as well as the use of a trickle feed protocol to optimize the quantification of protein digestibility by creating a plateau in the plasma ^13^C/^2^H enrichment ratio. This meant that at 150 min, when the first postprandial muscle biopsy was taken, participants had only consumed approximately 11.8 g of protein, which may not be sufficient to induce a maximal MPS response in older individuals, particularly after the consumption of blends containing a high proportion of plant protein, which would be expected to further limit EAA availability. Other potential limitations to this study include the fact that it was only conducted in older men in the age range of 65–75 with no medical conditions and a majority Caucasian population. This was done due to the relatively small sample size of the study, meaning that the study groups were made as homogenous as possible to limit potential sources of variability within the data. However, this does limit the applicability of the findings to other populations, including women, individuals of different ethnicities, younger individuals and clinical populations. This should be a focus of future research upon successful validation of the technique. Finally, all participants in the study received a 20 g trickle feed of their randomized protein blends. This was done to maximize the MPS response and to provide the protein blends in a commercially available dose. However, the accuracy and variability of future studies could potentially be improved by providing the protein doses based on body weight or lean body mass.

The similar performances of all four drinks for MPS could indicate that protein quantity is potentially the most important factor to maximizing the MPS response, more so than altering protein composition when consuming relatively high-quality protein sources. These findings agree with previous research assessing the role of blends of plant and animal sources, whereby no differences in MPS rates were reported between different blends despite subtle differences in plasma EAA and leucine concentration responses [48]. The present study extends these findings to older populations in the rested state, which is critical considering that older populations are most at risk of developing sarcopenia due to the inherent declines in muscle mass and function seen in ageing, as well as greater levels of inactivity in older populations [49]. Given the apparent need for older individuals to consume more protein [50], blends such as those assessed in the present study are likely to play a critical role in meeting these elevated requirements for healthy ageing.

## 5. Conclusions

This study has provided insights into the potential application of the dual stable isotope tracer technique to quantify protein digestibility, alongside measures of plasma amino acid concentrations and MPS, in vulnerable populations such as the elderly. Our findings suggest that the differing compositions of all four of the protein blends did not translate to any significant differences in the digestibility of the protein sources, as assessed by the dilution of the ^13^C/^2^H enrichment from the reporter spirulina protein and reference free amino acid mixture. Unsurprisingly, this translated to a lack of significant differences in postprandial plasma EAA concentrations and MPS responses to the four protein feeds. Notably, a combination of factors such as anabolic resistance in older individuals and certain study parameters, such as the need for a trickle feed approach and the relatively small sample size of the groups, may have further contributed to the lack of differences observed in this study. The dual stable isotope tracer technique is an extremely promising approach to the minimally invasive, routine quantification of protein digestibility, which still requires further development compared to historic techniques, particularly when looking to determine the digestibility of an unlabeled test protein against a labeled reporter and reference protein. Future work using this technique could enable the quantification of protein digestibility in a variety of food matrices and under a range of conditions, including assessing the impacts of different processing techniques on protein digestibility, as well as the effects of exercise training on protein digestibility measures, which are currently unexplored. There is also a need for the application of this technique to various populations not assessed in this study, such as women and individuals of a broader range of ethnicities, along with the critical application to clinical and vulnerable populations so as to optimize nutritional interventions.

## Figures and Tables

**Figure 2 nutrients-17-03328-f002:**
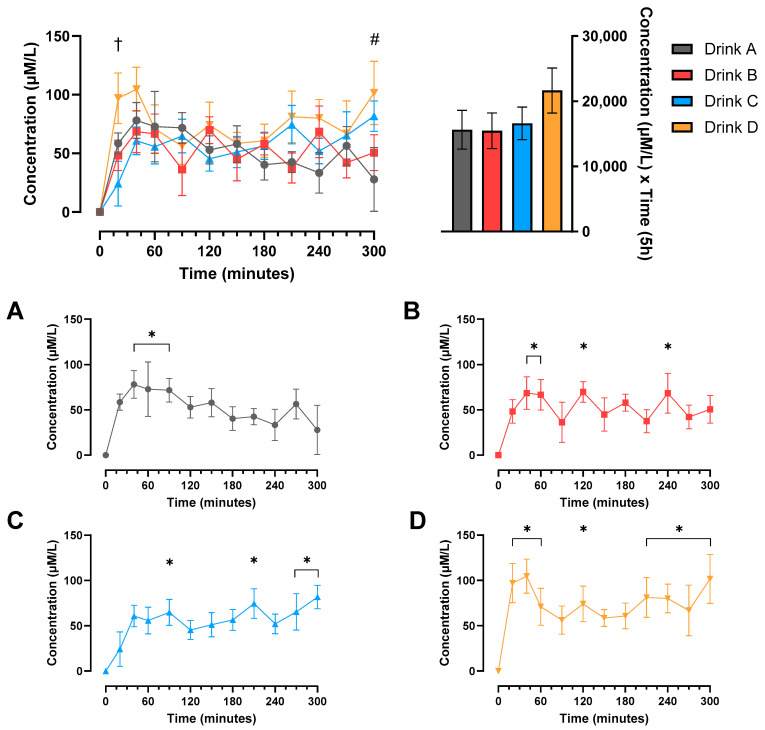
Change in plasma leucine concentrations compared to baseline over time and area under the curve (AUC) in older males. Values are mean ± SEM. Significant main treatment (*p* < 0.01) and time (*p* < 0.0001) effects observed. † Significant difference between drink D and drink C at time point (*p* < 0.05). # Significant difference between drink D and drink A at time point (*p* < 0.05). (**A**–**D**) The changes in plasma leucine concentrations compared to baseline in drinks A, B, C and D, respectively. * Significant difference from baseline at the time point (*p* < 0.05).

**Figure 3 nutrients-17-03328-f003:**
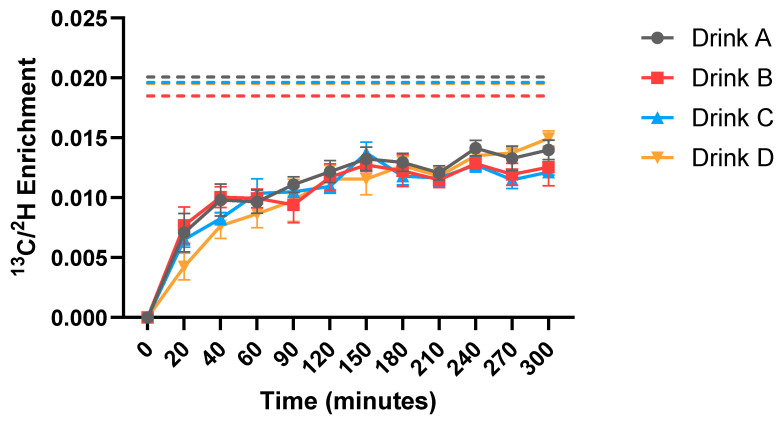
Change in plasma ^13^C/^2^H enrichment ratios (solid lines) and ^13^C/^2^H enrichment ratios measured in the respective drinks (dotted lines). Values are mean ± SEM. No significant differences between any of the drinks at any time points after reaching the plateau (all *p* > 0.05).

**Figure 4 nutrients-17-03328-f004:**
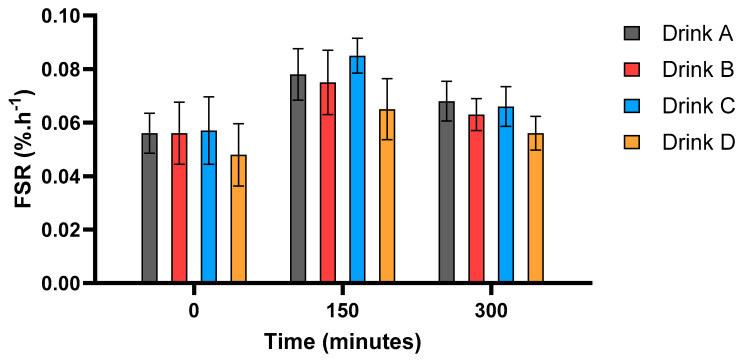
Myofibrillar protein synthesis (FSR, fractional synthesis rates, %·h^−1^) measured at baseline and 150 and 300 min post first feed in older males. Values are mean ± SEM. Significant main time effect (*p* < 0.01).

## Data Availability

The original contributions presented in this study are included in the article. Further inquiries can be directed to the corresponding author.

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
