# Peer review of "Investigating the Digestibility, Bioavailability and Utilization of Protein Blends in Older Adults Using a Dual Stable Isotope Tracer Technique"

_nutrients, 2025, doi:10.3390/nu17213328_

Round 1

Reviewer 1 Report

Comments and Suggestions for Authors

After reviewing the manuscript, I can say that the work is well planned. The literature used was well-selected. The research results support the stated aim of the work. However, some issues require clarification:

The abstract is too broad. It should be more concise.
In my opinion, the study group is not diverse, so the results may be unreliable.
Was appropriate physical activity taken into account for the project participants?

What was the general health status of the participants? Were laboratory tests performed?

What diet did the participants follow during the project? Why were only men included?

Was body composition analysis of the participants performed before and after the project?

Why weren't other methods of protein digestibility analysis used to compare the study results?

Reviewer 2 Report

Comments and Suggestions for Authors

This study is an interesting attempt to examine the digestibility, amino acid utilization, and muscle protein synthesis (MPS) responses of multiple protein blends in elderly subjects using dual stable isotope tracer techniques. While the methodology is sophisticated and the topic is of high clinical relevance, several concerns remain regarding the sample size and interpretation of the results.

1. Key findings include: The small sample size of eight patients per group may have lacked the statistical power to detect subtle differences in digestibility, blood amino acid concentrations, and MPS responses. The lack of significant differences between groups in this study may be due to insufficient statistical power rather than a true "no difference." We strongly encourage the authors to discuss this point more clearly and, if possible, provide post-hoc power calculations to explain whether the study was able to detect clinically meaningful differences.

2. Drinks B and C are described in the paper as having the same composition (35:25:20:20, whey/casein/soy/pea). If they are in fact identical, the reason for this should be clearly stated, and any misrepresentation should be corrected.

3. The annotations of statistically significant differences in Figures 1 and 2 (e.g., a, b, c, d) are duplicated, making it difficult to intuitively understand which groups correspond to which. We believe that clearer organization of the legends and annotations, and the addition of accurate P values ​​where possible, would make the figures easier to interpret.

4. The rapid increase in blood EAA concentrations 20 minutes after ingestion of Drink D (80% casein) is inconsistent with the properties of casein, which is generally considered to be "slowly digested and absorbed." While the authors offer some speculation, their argument would be more persuasive if they provided more in-depth interpretations or alternative hypotheses, such as the influence of amino acid composition or trickle-feeding.

Round 2

Reviewer 2 Report

Comments and Suggestions for Authors

The revised version includes clarification of the methodology and a more thorough explanation of the analytical procedures, significantly improving the transparency and reproducibility of the paper overall. The statistical analysis and limitations are more thoroughly explained, and the main concerns raised by the reviewers have been addressed with honest revisions. The overall flow of the text has also been improved, resulting in a clearer discussion.